# Early warning of trends in commercial wildlife trade through novel machine-learning analysis of patent filing

A. Hinsley [1,2] ✉, D. W. S. Challender[1,2], S. Masters [3], D. W. Macdonald[1], E. J. Milner-Gulland [1,2], J. Fraser [4,5] & J. Wright [2,6] ✉

Unsustainable wildlife trade imperils thousands of species, but efforts to identify and reduce these threats are hampered by rapidly evolving commercial markets. Businesses trading wildlife-derived products innovate to remain competitive, and the patents they file to protect their innovations also provide an early-warning of market shifts. Here, we develop a novel machine-learning approach to analyse patent-filing trends and apply it to patents filed from 1970-2020 related to six traded taxa that vary in trade legality, threat level, and use type: rhinoceroses, pangolins, bears, sturgeon, horseshoe crabs, and caterpillar fungus. We found 27,308 patents, showing 130% per-year increases, compared to a background rate of 104%. Innovation led to diversification, including new fertilizer products using illegal-to-trade rhinoceros horn, and novel farming methods for pangolins. Stricter regulation did not generally correlate with reduced patenting. Patents reveal how wildlife-related businesses predict, adapt to, and create market shifts, providing data to underpin proactive wildlife-trade management approaches.

The world is facing unprecedented rates of global biodiversity loss from threats including climate change, habitat loss and over-exploitation[1]. To reverse this biodiversity crisis, and achieve the goals of the post-2020 Global Biodiversity Framework, there is an urgent need to move away from reactive conservation towards proactive, evidence-informed action[1,2]. Wildlife trade involves a diverse array of species[3], and unsustainable trade has been linked to several hundred extinctions[4], as well as large-scale declines in species such as pangolins harvested for the medicinal and meat trade[5]. Beyond biodiversity impacts, effective wildlife trade management is an increasingly important global priority[6] due to links between wild animal markets and the origins of COVID-19[7,8], and wild animal welfare concerns in some markets, such as Asiatic black bears *Ursus thibetanus* farmed for their bile in Southeast Asia[9]. Adding further complexity, trade may also be essential to supporting livelihoods; in rural Nepal, harvesting of medicinal caterpillar fungus *Ophiocordyceps sinensis* may contribute 65% of income in some areas[10].

Despite the importance of robust management, priorities for policy and action to manage trade are often identified based on historical legal or illegal trade data with little proactive foresight, despite some approaches offering critical insights into emerging wildlife trade trends[11]. However, wildlife markets are constantly changing as entrepreneurs commercialise new species or products, such as rare python colour morphs[12], or expand markets for existing products, such as rebranding rhinoceros horn as a cancer treatment[13]. Furthermore, trade bans or other regulatory events may lead to the emergence of new substitute species from different regions, such as arapaima leather as a replacement for pangolin leather in the USA[14], or increased trade of African lion and South American jaguar products in Asian markets following the international commercial trade ban for tigers[15,16]. Farmed, artificially propagated, or lab-grown versions of a product may also emerge, such as synthetic rhinoceros horn[17]. Furthermore, businesses strategically adapt to shifting consumer or societal trends in product preferences, such as rapid shifts in the marketing of existing

[1]Department of Biology, University of Oxford, Oxford, UK. [2]Oxford Martin Programme on the Wildlife Trade, Oxford Martin School, Oxford, UK. [3]Naturalis Biodiversity Centre, Leiden, The Netherlands. [4]Saïd Business School, University of Oxford, Oxford, UK. [5]Imperial College Business School, Imperial College London, London, UK. [6]Oxford Internet Institute, University of Oxford, Oxford, UK. ✉e-mail: amy.hinsley@biology.ox.ac.uk; joss.wright@oii.ox.ac.uk

medicinal plant products to promote them as COVID-19 cures in early 2020 (e.g. *Artemisia annua* in Madagascar[18]). Data on how businesses create, adapt to, and predict changes in wildlife markets can provide important insights into future developments in the trade of different wildlife products.

Innovation provides novel ways to extract value from mature markets through the technological development of new products or processes, or the introduction of new business models[19,20]. Patent data provide a window into innovations in the commercial wildlife trade, as businesses are motivated to file patents to protect innovations and their commercial application from exploitation by others[21]. This protection lasts for up to 20 years, but patenting is recommended only in cases where the innovation concerned has significant commercial potential, due to the costs associated with obtaining and maintaining them[21]. Manual patent analysis has been used to draw conclusions on access and benefit sharing for marine biological resources[22,23], and to understand diversifying uses of wild orchids[24]. We propose to expand this approach through large-scale scraping and machine learning analysis of patent data, supporting the proactive management of wildlife trade towards sustainability. The analysis here draws on all patents filed between 1970 and 2020 relating to six commercially traded taxa, representing a range of product forms, geographies, and regulation types: bears, rhinoceroses, caterpillar fungi, pangolins, horseshoe crabs, and sturgeon (Table 1). We apply machine learning to detect trends in the topics of commercial innovation in this dataset over time, using Latent Dirichlet Allocation (LDA) topic modelling and Bayesian changepoint detection to identify points in the time series where wildlife patenting rate underwent fundamental changes, and to investigate links between patenting and exogenous events, such as the introduction of commercial trade bans.

We hypothesise that for actors in mature markets, such as traditional Chinese medicine, product innovation provides the opportunity to pursue a differentiation strategy and generate supernormal returns in a highly competitive space[25]. We further hypothesise that regulation affecting access to wild products (e.g., trade bans) will shape the type of innovation occurring. Theory suggests that actors either innovate to ensure they comply with new regulations, thereby achieving competitive advantage in the existing legal market[26,27], or become "avoidance entrepreneurs", aiming to avoid being subject to the regulations[28]. That it became legal to patent traditional Chinese medicine products in China only in 1993 suggests that concerted innovation efforts in these markets may be a recent phenomenon[29], with legislation now allowing for developments that would not previously have been patent-protected. Finally, we hypothesise that the changing nature of patents over time reveals how businesses view the future of wildlife markets. This may include increased patenting of new products to access different markets[24] or of currently illegal products that may soon become legal, as has been observed for marijuana patenting in the United States[30]. Our study period begins before CITES came into force in 1975, covering a time when global interest in wildlife trade increased rapidly, partly due to growth in Asian markets for species such as rhinoceroses and pangolins[6]. During this period global patent-filing also increased dramatically, driven by rapid technological advances and expansion of patenting to new sectors and countries[31]. We do not limit our analysis geographically but note that global patenting trends are affected by changes in domestic policy and support for patent-filing. In particular, changes in Chinese policy since the 1980s led to a significant rise in patenting, with China the most prolific patent-filing country since 2011[32], as well as a key wildlife consumer country.

## Results

### Substantial, and increasing, commercial innovation for both legal and illegal wildlife products

Commercial interest in key wildlife products has increased at a rate above background trends in patenting over the last half century. In total, 27,308 patents were filed for our focal taxa compared to 118,393,322 patents filed globally between 1970-2020. Horseshoe crab patents were the first to be filed in 1970, followed by sturgeon (1971), bear (1977), caterpillar fungus (1978), pangolin (1980), and rhinoceros (1988). Between July 1988, when patenting for all taxa was taking place, and December 2020, global patent filing showed a mean increase of 104%, while the mean increase in patent filing for our focal taxa was similar or higher (horseshoe crab: 102%; sturgeon: 129%; pangolin: 130%; bear: 115%; rhinoceros: 149%; caterpillar fungus: 143%). Patenting rates for all focal taxa increased from a median of 0 patents filed in 1970-1971, to a median of between 1 (rhino) and 46 (caterpillar fungus) per month in 2019-2020. Given that some of our focal taxa are globally threatened or illegal to trade, these findings represent a surprising level of commercial interest.

Despite the overall increasing trend, patenting rates increased significantly at taxon-specific points (Fig. 1). Key changepoints for bear (March 1992), caterpillar fungus (August 1992) and pangolin (August, 1995), which were all used in legal medicinal products at this time, corresponded with when China legalised traditional Chinese medicine patenting in 1993[29]. Medicinal rhinoceros products were banned in China in 1993[33], with the most significant changepoint (March 2008) corresponding with 2008's significant increase in rhinoceros poaching[34]. The largest shift in horseshoe crab patenting rates (August 1987) coincided with the US Food and Drug Administration issuing key guidance for the use of Limulus Amebocyte Lysate (LAL) [35]. Sturgeon patenting rates shifted most notably (April 2003) following a 2001 temporary ban on Caspian sea caviar exports, and a 2002 CITES Resolution on measures to conserve sturgeon stocks[36].

Monthly patenting rates peaked between 2013 and 2017 for all taxa, first for rhinoceros (53 filed in January 2013) and pangolin (150, December 2013) then caterpillar fungus and bear (485 and 34 respectively, December 2015), sturgeon (117, April 2016), and horseshoe crab (13, September 2017). Peak patenting rates for rhinoceroses occurred in the year with the highest recorded illegal rhinoceros hunting[34], while peak pangolin patenting coincided with the movement of pangolin species to higher IUCN Red List categories of extinction risk[37].

### Wildlife trade bans do not necessarily reduce commercial innovation

In most cases, patenting rates continued to increase following regulatory events that banned or heavily restricted trade internationally, or in key trade countries: all 526 patents for rhinoceros were filed after a 1977 international commercial trade ban for wild products, with 476 of these focussed on products containing rhinoceros horn rather than innovations in producing synthetic or farmed alternatives. Of these, 426 patents were filed in China after the domestic trade ban in 1993. This implies that companies are either banking on a relaxation of legislation banning the rhinoceros horn trade, or planning to commercialise products using illegally acquired horn. Similarly, increasing patenting rates for pangolins continued following CITES zero export quotas for some species in 2000. Increases in both sturgeon (Fig. 2a) and caterpillar fungus patenting rates followed national-level bans in key harvesting areas (Caspian Sea[36] and Nepal[10] respectively). The exception is the 2017 CITES Appendix I listing of pangolins, after which patenting rates, which had been increasing rapidly since the mid-2000s, decreased from a median of 81 patents per month in 2014 to a median of 31 per month in 2017 (Fig. 2b).

### Wildlife markets are diversifying

Innovation has led to a diversification of wildlife-derived products over time (Fig. 3), including the emergence of novel product types such as rhinoceros horn snuff products, and livestock feed containing pangolin scales (Table 2). However, in all cases, the number of new patents related to established uses, such as medicines using bear bile,

**Table 1 | Details of the markets, geographical locations of trade, and regulations applied to the trade in the six focal taxa used in the study**

| Taxon | Primary products | Primary market | Location of harvest | Location of market | Regulations |
|---|---|---|---|---|---|
| Caterpillar fungus (primarily *Ophiocordyceps sinensis* and *Cordyceps militaris*) | Whole organism | Traditional medicine; Food | Nepal, Bhutan, China, India | Global, particularly in Asia | Not CITES listed. Several local and national restrictions on harvesting. |
| Pangolin (Manidae family) | Scales; meat | Traditional medicine (scales); food (meat) | Several countries in Africa and Asia | Primarily China and Southeast Asia. Also, sub-Saharan Africa | All listed in CITES. App. I in 2017. Harvesting and trade bans in virtually all range states. |
| Horseshoe crab (Limulidae family) | Blood; whole organism | Biomedicine (blood); fishing bait | Primarily USA | Global (blood), USA (bait) | Not CITES listed. Several local and national restrictions on harvesting. |
| Sturgeon (Acipenseridae family) | Eggs; meat | Food | Primarily Caspian and Black Seas | Global, particularly Europe and USA. | All listed in CITES App. II in 1998, one species in App. I in 1975. National and regional restrictions on fishing. |
| Bears (primarily *Ursus thibetanus* and *U. arctos*) | Bile | Traditional medicine | Southeast and East Asia. | China and Southeast Asia | *U. thibetanus* listed in CITES App. I in 1979. Some *U. arctos* populations listed in App. I, and all other species in App. II in 1992. Legal bile farming in different Asian countries between 1980 and present. National restrictions on harvest and trade in several countries. |
| Rhinoceros (Rhinocerotidae family) | Horn | Traditional medicine | Several countries in Africa and Asia | China and Southeast Asia | All listed in CITES App I in 1975, some populations in App II. National restrictions on trade. Several proposals to legalise domestic markets. |

continued to exceed those for novel product types. Some novel products showed sudden increases in patenting, such as 50 patents for rhinoceros horn snuff filed in 2013, while others show gradual increases in the proportions of patents filed including applications of horseshoe crab blood in the electronics industry in the 1980s, which became an increasing proportion of horseshoe crab patents from the mid-2000s onwards.

All taxa showed diversification in product source, including product and process innovations focussed on methods to artificially breed, cultivate or farm individuals of focal taxa (hereafter referred to as 'farming'), synthetic versions of wild products, or alternatives made from domestic species. Farming innovations included methods for feeding (e.g., pangolin cub feed), breeding (e.g., sturgeon insemination), increasing yield (e.g., inoculating insect larvae with caterpillar fungus), and harvesting products from live animals (e.g., rhinoceros horn 'scraping' tools). Synthetics patents proposed novel methods for synthesising active ingredients in wild products (e.g., recombinant Factor C to replace horseshoe crab blood), or using domestic animal genetic material or final products to replace wildlife-based ingredients (e.g., chicken bile to replace bear bile). There were more farming than synthetic-related patents filed for sturgeon (farming: $n = 1190$; synthetics: $n = 90$) and caterpillar fungus (farming: $n = 4968$; synthetics: $n = 2412$); both taxa are legally traded on a large scale but wild population declines or local harvesting regulation restrict access to wild products. Bear farming patents first emerged in the 1980s and were filed in smaller numbers than synthetic-related patents during our study period (farming: $n = 59$; synthetics: $n = 273$). More synthetic than farming-related patents were filed for horseshoe crab (farming: $n = 90$; synthetics: $n = 163$). In contrast, more rhino-farming patents were filed than those for synthetic alternatives (farming: $n = 41$; synthetics: $n = 27$). While more synthetic-related pangolin patents ($n = 113$) were filed than those related to farming ($n = 44$), the filing rate of farming patents increased coinciding with the CITES Appendix I listing announcement in 2016. In addition to product patenting, patents for the detection of counterfeit or synthetic wildlife products also increased over the study period.

## Discussion

Our analyses show dramatic increases in commercial interest across a diverse range of wildlife-related products over the past 50 years, including in threatened and illegally traded species. Patenting is increasing worldwide, in all sectors, but the above-average growth of patent-filing for wildlife-related products demonstrates active innovation[24,31], despite increases in regulatory and non-regulatory efforts to reduce unsustainable trade[6]. Most wildlife-related patents in our study focussed on expanding the use of wildlife to new products or industries, with only 9470 patents during our study period (34.6% of patents for all taxa) referencing innovations in farmed or synthetic sourcing.

Patent data provide new evidence to support actions that anticipate future developments in unsustainable harvesting by identifying emerging trends in wildlife use: the continued filing of product patents for taxa subject to trade bans, such as the medicinal and agricultural products derived from pangolin scales, indicate business confidence in future reopening of these markets. Patent-filing trends in processes to produce farmed or synthetic alternatives, such as farming methods for rhinoceroses, reveal commercial perceptions of the future of markets in which wild products are no longer legal to trade. While business predictions may be inaccurate, they represent the intent of commercial entities, and as such patents are a valuable data source for monitoring interventions aiming to affect commercial decisions in the long term.

As hypothesised, wildlife markets are a focus of constant innovation. Product differentiation occurred across our focal taxa, including the recent emergence of novel product types such as snuff or

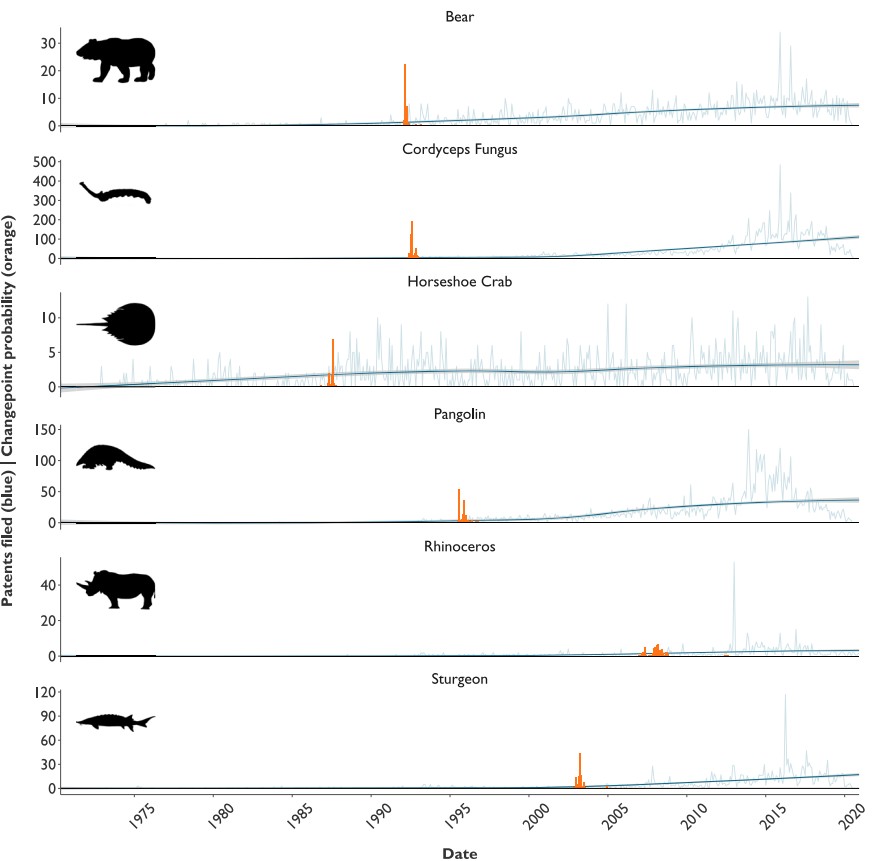

**Fig. 1 | Trends in patent filing for six taxa over 50 years.** Patent filings per month (blue, actual values and smoothed average shown); and probability of changepoint (orange) for bear, caterpillar fungus (cordyceps), horseshoe crab, pangolin, rhinoceros, and sturgeon between 1970 and 2020. Silhouettes are public domain, from https://www.phylopic.org/., except Cordyceps, which is adapted from original photograph at https://commons.wikimedia.org/wiki/File:Cordyceps_Sinensis.jpg uploaded by William Rafti. The original image is licensed under the Creative Commons Attribution 2.5 Generic license: https://en.wikipedia.org/wiki/en:Creative_Commons.

fertiliser containing rhinoceros horn. Currently, novel wildlife products are detected only after they have been commercialised, most often after being seized from illegal trade[16] or recorded for sale in real world or online marketplaces[38]. By contrast, inspection of patent-filing trends reveals emerging products that may not come to market for several years, due to lengthy patenting processes and the long-term nature of patent protection[21]. Trend data can therefore inform enforcement officers of new wildlife-derived products, or can inform interventions to change consumer or industry behaviour in sectors in which wildlife products have been less common, before these behaviours become prevalent and ingrained. As a concrete example, conservation efforts currently focus on the use of pangolin scales in traditional medicine products; emerging patenting trends suggest that attention should be drawn to future pangolin use in the agricultural feed and fertiliser industries.

In-depth studies of patenting for specific taxa, using the methods employed in this work, provides a characterisation of the breadth of products available in markets involving these taxa[24]. These should be conducted periodically for priority taxa to identify emerging products and threats. For identifying emerging threats more broadly, there are two potential extensions to these approaches. Firstly, automated searches could be conducted for a wide range of species, identified via some method of prioritisation, for example, the IUCN Red List or expert advice on taxa that may be important to monitor. Then, if increased rates of patenting were detected, these could be investigated in greater detail, to characterise patent types and emerging markets. Alternatively, searches based on keywords that characterise particular classes of patents, such as Traditional Chinese Medicine,

could be monitored for the emergence of new species in patent documents. This would reveal the potential for the inclusion of previously unexploited taxa in new products and applications. Of particular importance for the detection of emerging patenting behaviour would be periods following major shocks that affect entrepreneurship (e.g., through opening new opportunities, closing existing markets, or changing consumer attitudes and behaviour), such as changes in regulatory regimes or global events, such as the COVID-19 pandemic.

We found no correlation between regulations that banned or restricted wild product trade and changes in patent-filing behaviour. Past research has shown that the impact of wildlife trade restrictions is often unpredictable, with CITES listings increasing trade in the short term[11], and rapidly shifting exploitation to other, less regulated species[14]. Patent data support this, with pangolin patenting rates decreasing following CITES Appendix I listing but rhinoceros and other taxa showing increased patenting rates following similar bans and restrictions. By contrast, changes in broader industry regulations that did not focus on wildlife specifically, such as removal of restrictions around patenting traditional medicine products in China[29], coincided with strong shifts in patent focus and filing rates across several taxa. Lack of awareness or understanding of CITES rules within industry may be to blame, and communicating regulations and penalties related to use of different wildlife products to companies patenting products related to them may begin to address this. While there is often no formal restriction on the patenting of illegal products, patent offices could flag filings for an innovation that relies on banned species; this would offer a direct means for government Wildlife Departments to engage with commercial entities on wildlife usage.

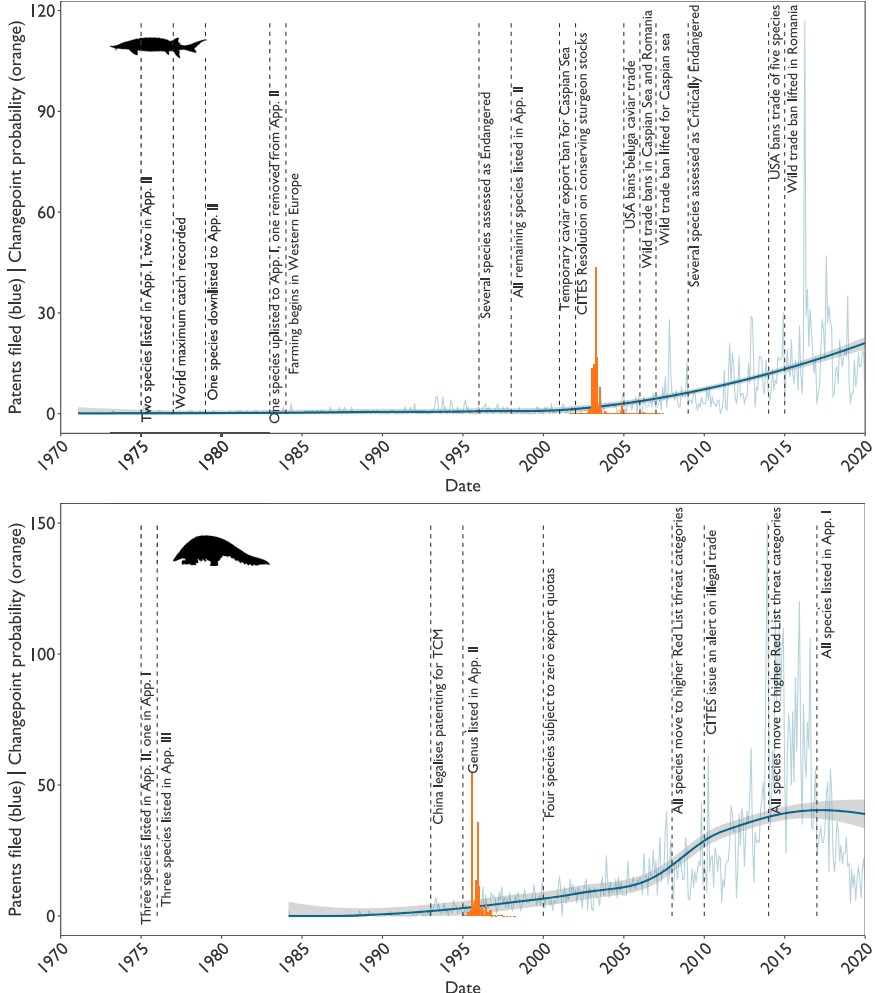

**Fig. 2 | Patent filing trends and key events for sturgeon and pangolin.** Patent filing trends for pangolin and sturgeon alongside key events related to those taxa. Patent filings per month (blue, actual values and smoothed average shown); and probability of changepoint (orange) for patenting of sturgeon (top) and pangolin (bottom) between 1970 and 2020, showing key regulatory and nonregulatory events relevant to these taxa that occurred during this period. Silhouettes are public domain, from https://www.phylopic.org/.

The sustained commercial interest in illegal products shown here reveals the extent of business confidence in the potential of future markets for these products. Increased patenting of rhinoceros products detected in our analysis, even after CITES and national-level bans, may have resulted from legal uncertainty relating to multiple proposals for the re-opening of legal markets[33]. Such uncertainty provides opportunities for entrepreneurship as businesses stake their claim in potentially lucrative emerging markets[30]. By contrast, rapidly declining wild populations, and subsequent strong support for associated CITES Appendix I listings, may have reduced uncertainty surrounding the future of wild pangolin trade, leading to the decline in commercial interest we observed following the CITES uplisting. While overall pangolin patenting decreased after this event, farming-related patents increased. Where wildlife trade bans are implemented, ensuring that these bans are seen to be long-term for taxa for which wild-sourced trade is clearly unsustainable would reduce uncertainty, thereby stimulating innovation related to farmed or synthetic alternatives. Scanning the patent data for species that have recently been subject to trade bans or other regulations to restrict trade could underpin targeted work with businesses or sectors to determine the reasons why they want to patent an illegal product, the best interventions to ensure that this commercial interest is sustainable.

If wildlife markets based on rare wild taxa are to continue, they must ensure sustainable sourcing, such as well-managed medicinal plant harvest[39], or shifts to farmed, synthetic or domestic alternatives, such as rhinoceros horn made from horse hair[40]. While a broad shift from wild to captive-bred CITES trade has occurred over time[41], introducing new, farmed or synthetic alternatives to a market can be challenging, with no guarantee of success, especially if the wild alternative is cheaper, or preferred by consumers[42]. While alternatives may be proposed by NGOs or other actors based on conservation or welfare goals, businesses will only adopt them if they are likely to be competitive and profitable. Working closely with the businesses who are patenting these products should be a priority, to ensure that novel alternative products meet conservation, animal welfare, and commercial goals[17].

The wildlife trade operates much as any other commercial sector, and patent data provide significant insights into how businesses view the future of wildlife markets. We note that not all traded wildlife will be the subject of patenting, but patent data should be triangulated with other data sources to add an additional layer of information about commercial trade in certain markets that has so far been overlooked. Furthermore, not all patented products will be commercialised, but the combination of patent data with ground-truth information drawn from businesses and consumers provides a means to prioritise those products and sectors which could be the focus of proactive interventions. Ongoing scans of patent data can provide information regarding emerging trends, such as where commercial interest in wild taxa is

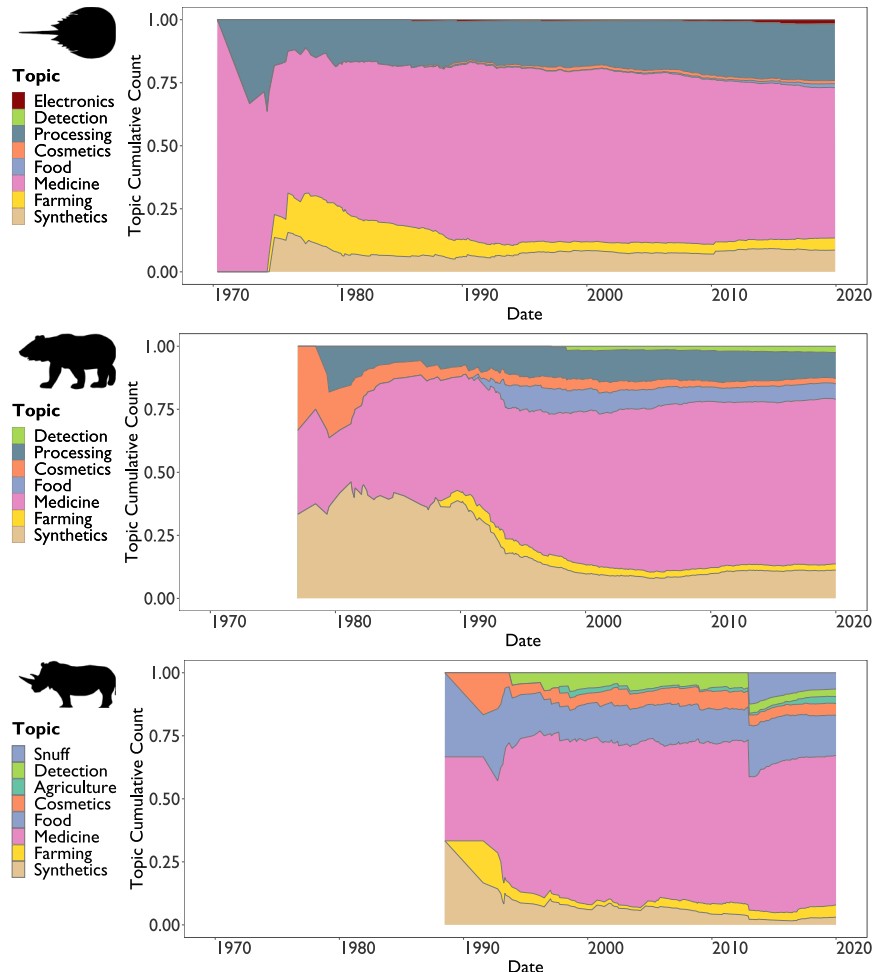

**Fig. 3 | Diversification of commercial interest in six taxa over time.** Cumulative proportion of horseshoe crab, bear, and rhinoceros-related patents between 1970 and 2020, showing diversification of labelled topics over time. Topics are not mutually exclusive: total count represents all labels applied to patents rather than all unique patents. Silhouettes are public domain, from https://www.phylopic.org/.

increasing, while also identifying novel product types so that their effect on wild population sustainability can be assessed ahead of time. With extinction risk increasing at unprecedented rates, and exploitation for trade threatening countless species, action is urgently needed to ensure long-term sustainability of these markets. Working closely with businesses and entrepreneurs involved in the wildlife trade, who can be easily identified from patent-filing data, is essential to co-design proactive measures that will align the goals of conservation with commercial innovation. Prioritising interventions that encourage innovation in sustainable or alternative sourcing for products derived from threatened taxa, rather than seeking to remove innovation in these markets, presents a means to harness commercial forces towards sustainability.

## Methods

### Identifying and refining case studies

We identified a long-list of wild-type taxa (whether wild-sourced or farmed), traded commercially in more than one country, which had been subject to regulatory or non-regulatory measures designed to change trade in some way. Long-listed taxa were pangolins, bears, rhinos, freshwater turtles, crocodiles, snowdrops, horseshoe crabs, devil's claw plants, caterpillar fungus, and sharks. We then produced a final shortlist of taxa that covered a variety of trade purposes and trade control measures: bears traded primarily for their bile used in medicine (family Ursidae); caterpillar fungus used in medicine and health products (genera *Cordyceps* and *Ophiocordyceps*); horseshoe crabs (family Limulidae); pangolins used in medicine and for food in different regions (family Manidae); rhinos traded primarily for their horn, used in medicine or ornamentally (family Rhinocerotidae); and sturgeon primarily traded for their caviar for the food trade (family Acipenseridae). These include cases where commercial trade is restricted to regional markets (e.g., bear bile traded primarily for traditional medicines in China and Southeast Asian countries), as well as those with global markets (e.g., horseshoe crab blood harvested and traded worldwide for the biomedical pharmaceutical industry). We selected three examples that are subject to international commercial trade bans for all species (bears, rhinos, pangolins), as well as national-level restrictions on wild harvest and trade in all or most range states. In addition, we selected three examples with local and national restrictions on harvest, which differ between range states and species (sturgeon, caterpillar fungus, horseshoe crab). Four of our examples have a current legal commercial trade in farmed or synthetic alternatives (bear, horseshoe crab, sturgeon, caterpillar fungus). All examples have shown some change in their regulation, or the commercial availability of alternative products in our study period.

### Selecting keywords

Patent applications must provide detailed information on the component parts or processes of an innovation, background information on the context in which it could be used, or what it is designed to

**Table 2 | Examples of patents filed during our study period for each taxon-topic combination (as identified in the LDA and manual topic analysis), excluding topics with very small numbers of patents**

| Taxon | Indicative patent topics (example patent in topic, filing year, country) |
|---|---|
| Bear | Medicine (Preparation of cough medicine containing bear's bile, 1990, China); Food (Bear gall beverage, 2004); Cosmetics (Toothpaste containing bear gall microcapsule, 2019, China); Synthetics (A kind of artificial bear gall powder and preparation method thereof, 2014, China); Farming (Adult black bear compound feed for improving quality of bear gall powder and preparation method thereof, 2013, China); Processing (Preparation method for bear bile powder, 2015, China); Detection (Method for identifying truth and false of bear bile powder by polymerase chain reaction, 2011, China). |
| Caterpillar fungus | Medicine (Cordyceps sinensis-containing traditional Chinese medicine composition for dispelling effects of alcohol and protecting liver and application thereof, 2016, China); Food (Superfine cordyceps food and its production process, 2000, China); Synthetics (Method for producing cordyceps sinensis powder by virtue of liquid fermentation on mixed bacterial strains, 2013, China); Farming (Method for producing cordyceps sinensis using larvaes of fly, 2003, South Korea); Detection (A kind of detection method of certified products Cordyceps, 2013, China). |
| Horseshoe crab | Medicine (Method of detecting an endotoxin using limulus amebocyte lysate substantially free of coagulogen, 2017, 10 applications [priority: USA]); Synthetics (Method for recombinant production of horseshoe crab Factor C protein in protozoa, 2019, 11 applications [priority: Europe]), Farming (Artificial breeding and culturing method for horseshoe crabs, 2011, China); Electronics (Semiconductor devices including a support for an electrode and methods of forming semiconductor devices including a support for an electrode, 2012, South Korea, USA, China, Japan); Cosmetics (Beautifying and skin repairing sodium hyaluronate gel coated with stem cell complex factor, 2019, China). |
| Pangolin | Medicine (Health-care trousers capable of being worn with oneself, 2019, China); Cancer medicine (Medicinal preparation for treatment of tumour, 1995, China); Food (Lactation-promoting nutritional soup and preparation method thereof, 2013, China); Synthetics (A kind of artificial pangolin compound powder, 2018, China), Farming (Pangolin cub feed, 2014), Processing (A kind of concocting method of raw pangolin, 2017, China); Agriculture (Female Huzhou sheep feed used in milk production period and preparation method of female Huzhou sheep feed, 2014, China); Cosmetics (Spot-removing antiwrinkle cream, 2001, China). |
| Rhinoceros | Medicine (Medicine for treating blood diseases 2016, China); Food (Medicated food for adjuvant therapy on cancer, 2013, China); Synthetics (Synthetic rhinoceros horn analogues, 2014, World, Europe, USA, China), Farming (Self-suction living rhinoceros horn scraping tool, 2008, China); Detection (Molecular identification method of authenticity and category of rhinoceros horn products, 2012, China); Agriculture (Non-residual natural environment-friendly biological pesticide, 2013, China); Snuff (Rosemary scented snuff, 2013, China). |
| Sturgeon | Medicine (Medication for impotence containing lyophilised roe and a powdered extract of Ginkgo biloba, 1996, USA); Food (Process for the production of an alcoholic beverage using caviar, 2010, UK, France, Russia); Synthetics (Process for obtaining sturgeon caviar analogue, and product thus obtained, 1997, Europe, Japan, USA, World), Farming (Method of cage growing of market sturgeon species at early stages of ontogeny, 2012, Russia); Leather (Environment-friendly leather-producing process by using sturgeon skins, 2013, China). |

replace. Our focus was on patenting for products that contained parts of our focal taxa, or methods designed specifically to change the production or processing of these products. We therefore assumed, based on previous studies (Masters et al. 2020), that the taxa binomial or common name would be included in relevant patents, even those for products designed to act as an alternative to the wild products (e.g., a bear bile replacement made from chicken bile). We therefore identified initial keywords related to the Latin binomial or common name of each product in English, and in any key languages relevant to the primary areas of trade or consumption (e.g., 'pangolin', 'Manis' and the Chinese pinyin translation 'chuan shan jia'). We used these keywords to scrape the Google patents database and then randomly selected 200 patents from the results, which <redacted> used to manually check and refine keywords. This involved identifying patents containing those keywords that were not relevant to our case studies (e.g., 'pangolin' is used as a term in *Drosophila* genetic research, meaning that multiple patents containing this word were not focused on pangolins), and identifying any keywords associated with relevant patents that we had not included. We also excluded certain keywords that, whilst accurate, did not uniquely match a significant number of patents and also led either to translation errors or high rates of false positives (Table 3).

**Patent scraping**

To assess the growth and development of commercial interest in wildlife-derived products, and in particular to identify emerging topics and concepts, we used large-scale analysis of patent filing data. Patents represent the leading edge of technological development by corporations seeking to dominate and drive an industry, responding to perceived or identified needs in the marketplace. As patent filing is necessarily public, patent data provides a convenient and open data source by which to study emerging trends in a range of industries as well as the key concerns that are present in the text of existing patent filing behaviour.

To understand the landscape of patenting of wildlife-derived products, we scraped public patent data via the Google Patents search engine. This tool allows keyword-based search in the text and metadata of patents, with search terms automatically translated into multiple languages. We used the open-source browser automation library selenium, and specifically its R interface rselenium[40] to obtain a full list of patents matching a range of identified search terms of interest for given taxa.

The resulting full list of patents returned by the Google Patents search engine was then used to scrape patent text directly. This resulted in a dataset of all filed patents matching the provided search terms, including the names of the filing entities, the patent authors, the dates of first filing of a patent ('priority date'), the text of the patent abstract, and the full description of the associated invention. For our initial set of search terms, the total dataset comprised 27,308 patents across 35 search terms for six taxa.

Once downloaded, the patent data was subjected to initial filtering and processing to remove duplicate patents and irrelevant confounding search terms. Due to the nature of the data, it is difficult a priori to ensure that search terms do not produce spurious irrelevant results, and manual filtering to validate the downloaded data is required.

With a scraped and filtered dataset, we first identified the trends in the patent filing rate over time. To calculate the mean increase in patent-filing rate for the global patents across all sectors and our focal taxa, we started in 1985, the first year for which every taxon has had at least one patent filed and finished in December 2020. We calculated the year-on-year percentage change (January-December). Ignoring any years that had zero patents, we calculated the mean percentage change in patent filing across all years.

**Filing frequency changepoint analysis**

A key area of interest in patent filing behaviour is the rate at which patents for a given technology are filed. More specifically, we were interested in moments at which a *changepoint* occurs in patent filing, representing a significant shift in the underlying rate at which corporations file patents relating to a given concept.

We conducted a Bayesian changepoint analysis[41] on the rate of monthly patent filing for a given set of patent search results. This

**Table 3 | Final keywords used to scrape the Google Patents database for patents related to our focal taxa**

| Taxa | Keywords | Notes |
|---|---|---|
| Bear | 'bear bile', 'bear farm', bear 'gall bladder', bear 'gall powder', bear gall-bladder, 'fel ursi', 'ursodeoxycholic', 'ursus arctos', 'ursus thibetanus', 'xiongdan', '熊胆'<br>**NOT** 'ma huang jia zhu tang' | Our keywords focussed on bear bile as the key product in trade because general keywords, such as 'bear' resulted in a lot of false positives. Ma huang jia zhu tang was sometimes mistranslated as 'bear grass'. |
| Caterpillar fungus | 'cordyceps', 'caterpillar fungus', 'aweto', 'dongchongxiacao', 'dong chong xia cao' | Other names for Cordyceps (e.g., Yarsagumba) did not uniquely match any patents. |
| Horseshoe Crab | 'horseshoe crab', polyphemus, 'tachypleus tridentatus', 'tachypleus gigas', 'carcinoscorpius rotundicauda', limulus<br>**NOT** 'soft-shell' | Some patents for soft-shell crabs were mistranslated as 'soft-shell horseshoe crab' |
| Pangolin | 'pangolin', 'squama manis', 'jia zhu', 'pao shan jia', 'chuan shan jia', 'squama manitis', '穿山甲', '醋山甲'<br>**NOT** 'drosophila' | *Pangolin* is the common name of a fruit fly gene and occurs frequently in *Drosophila* genetic research. |
| Rhinoceros | 'rhinoceros', 'rhino', 'diceros', 'ceratotherium', 'dicerorhinus'<br>**NOT** 'rhinoceros beetle', 'oryctes', 'polyporus rhinoceros', 'giraffe rhinoceros', 'game', 'toy', 'software' | *Oryctes rhinoceros* and other rhinoceros beetles are a common agricultural pest. Many patents for rhinoceros were games or toys with rhinoceros characters. |
| Sturgeon | 'sturgeon', 'acipenser' | Other sturgeon genera names (e.g., *Huso*) were not found independently in the patents data, and inclusion led to high rates of translation errors. |

approach models the rate of patent filing in each month as drawn from a probability distribution, specifically a negative binomial distribution allowing for flexible description of count data. To identify a changepoint, we allowed the data to be described by two separate parameterisations of a negative binomial distribution – one prior to a potential changepoint, and one after. The changepoint is identified as a parameter describing a time point at which the two separate instantiations of the negative binomial model the observed data optimally.

We fitted the changepoint model using the Stan probabilistic programming language[42], through its R interface cmdstanr. The Bayesian changepoint provides not only a point estimate of the changepoint, but a probability distribution across potential changepoints, allowing us to easily assess the uncertainty in the models. While the approach taken here focuses on a single changepoint, it is possible to segment the dataset at identified changepoints to identify subsequent changepoints of interest.

Using Stan's default Hamiltonian Monte Carlo sampler (HMC), all changepoint models were conducted with the following parameters:

| | |
|---|---|
| Likelihood | Negative binomial, alternative parameterisation as detailed at: https://mc-stan.org/docs/functions-reference/unbounded_discrete_distributions.html#nbalt |
| Prior for $\mu$ (early and late) | 10 |
| Prior for $\varphi$ (early and late) | 1 |
| Chain iterations | 2000 |
| Number of chains | 4 |
| adapt_delta | 0.95 |

All chains were assessed for appropriate mixing across chains and convergence using standard approaches for Stan models: both visual inspection of traceplots and assessment of appropriate R-hat statistic for convergence.

For this work, we focused on identifying a key changepoint in the model, which is presented alongside a timeline of identified key real-world events relevant to legislation, regulation, or public attention drawn to a given taxon. While this approach cannot draw causative inferences between observed patent filing behaviour and exogenous

events, it places shifts in patent filing in their surrounding social, technological, and political context.

### Topic modelling

The changepoint analysis serves to identify the timeframe of key shifts in patent behaviour but does not interrogate the content of the patents. Latent Dirichlet Allocation (LDA) topic modelling is a widely used natural language processing (NLP) technique applied to identify commonly co-occurring themes in a large corpus of data. Topic modelling treats each document in a corpus as resulting from an underlying probability distribution of words, and splits the corpus into a defined number of separate 'topic' probability distributions. Each document can then be classified according to the given topic distribution that best describes its content.

The result of applying this approach is that a large corpus of documents can be automatically classified into sets of related documents, according to the key terms that appear in them. The characteristic terms for each topic, then, can be identified to allow human-identified semantic understanding of the classifications. In this sense, topic modelling as an unsupervised machine learning technique is best considered an automated aid allowing large-scale human-guided classification and understanding of large corpora of documents.

For the purposes of this analysis, topic modelling allowed us not only to classify related documents together, but to identify key themes and terms that run through the patent filing dataset, and to determine whether certain terms are emerging, growing, or falling in popularity. In the analysis presented here, we focus on the text of patent abstracts, rather than the full description text, as this provides a more focused view of the major themes of a patent, rather than the detailed technical description that is found in the full description text.

Topic number selection was achieved through balancing of the relative semantic coherence and exclusivity of the resulting models.

### Thematic coding for manual topic identification

We used the topics identified during the LDA topic modelling phase, combined with a codebook based on the Economic Botany Data Collection Standard (https://www.tdwg.org/standards/economic-botany/) to define broad topics and more specific subcategories related either to the use of the species in a novel product (medicine, food, agriculture, cosmetics) or to an innovative process associated with the trade of the taxa (production [e.g. farming, cultivation], preparation [e.g. extraction of active ingredients], detection [e.g. genetic differentiation of similar species]).

We defined keywords for each topic using the first five non-generic terms of each LDA-identified topic, and keywords associated with the Economic Botany standards (e.g., 'disorders' or 'pain' to cover various medical applications). We then labelled all patents containing these keywords as belonging to at least one topic; a patent could be labelled as relating to two topics if, for example, it related to a new medicinal product ('medicine') using a synthetic version of the wild taxa ('synthetic alternative'). We then used an iterative process, where we manually reviewed all labelled and unlabelled patents to identify missing keywords relating to our existing topics, identify new topics that had not yet been found, and highlight mislabelled patents. We repeated this process until all patents were labelled with at least one topic (see Appendix 5 for all topics and keywords).

## Reporting summary

Further information on research design is available in the Nature Portfolio Reporting Summary linked to this article.

## Data availability

The data that support the findings of this study are not openly available due to all patents being the copyright of the filing entities – data are available from the corresponding author upon reasonable request.

## Code availability

All code was written in GNU R, for data processing, and Stan for statistical modelling. The code is released under a Creative Commons Attribution-ShareAlike 4.0 International licence. (CC BY-SA 4.0). Code is deposited in Oxford's Open Research Archive and can be found at https://oxris.ox.ac.uk/viewobject.html?cid=1&id=2001501.

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

## Acknowledgements

AH was supported by the Kadas Senior Research Fellowship at Worcester College, Oxford, and the Oxford Martin Programme on the Wildlife Trade. EJMG and DWSC acknowledge funding from the UK Research and Innovation's Global Challenges Research Fund (UKRI GCRF) through the Trade, Development and the Environment Hub project (project number ES/S008160/1).

## Author contributions

A.H., J.W., D.W.S.C. and S.M. contributed to the conception, design, and initial data filtering. J.W. wrote the code to acquire and clean data, and perform the statistical analysis. A.H. and J.W. analysed the data. A.H. and J.W. drafted the paper, with significant input and comments from D.W.S.C., S.M., D.W.M., E.J.M.G. and J.F. to produce the final draft. A.H. and J.W. contributed equally to this work and should be considered joint first and co-corresponding authors.

## Competing interests

The authors declare no competing interests.
