## [Peer Review File · Nature Communications]

Early-warning of trends in commercial wildlife trade through novel machine-learning analysis of patent filingReviewers' Comments:

Reviewer #1:

Remarks to the Author:

This is a fascinating study on an previously underappreciated component of the wildlife trade. I do wonder how effective patents will be as an early warning for emerging threats to novel wildlife markets. Presumably the number of wildlife species with patents is a very small proportion of all wildlife traded - and a comprehensive understanding of which species are patented and at what stage in their wildlife use/exploitation would be useful. I would like the authors to add some discussion about how emerging patents would be detected for novel species - rather than simply the retrospective analysis as currently provided.

Reference 14 would be better as: Heinrich, S., et al., 2019. Of cowboys, fish, and pangolins: US trade in exotic leather. *Conservation Science and Practice*, 1(8), p.e75.

Line 40. 'Non-wild' is an unclear term, are you referring to both farmed and artificial? It would be good to better define this.

Line 154. Given that 'the number of new patents related to established uses, such as medicines using bear bile, continued to exceed those for novel product types' - how do the authors consider their general findings will apply to novel less consumed/marketable species?

Reviewer #2:

None

Reviewer #3:

Remarks to the Author:

I enjoyed this paper a lot, and learned from it. Thank you for that.

I have very minor suggestions:

Ln 43: owl trade in Indonesia due to their rising popularity as pets following the Harry Potter films¹⁸. Check the validity of this claim, as I have read that it is controversial and more anecdotal than anything else.

Ln 56: remove last comma

Ln 131: For me, this is the most important finding of the paper and warrants more discussion and clarification. At least I would like to read more on this trend.

Many thanks for the reviews of our paper. We are very grateful for the reviewers insights and advice on how to strengthen the paper, and we have addressed their comments below. In addition to the edits in response to reviewer comments, we have also replaced some references in relation to the innovation and emerging markets points we make – this is so that we can ensure we are using the most up to date references. We have uploaded both a tracked changes and clean version of the MS so that our changes can be viewed easily.

Do not hesitate to contact us if any further edits are needed.

Response to Reviewers

REVIEWERS' COMMENTS

Reviewer #1 (Remarks to the Author):

Comment 1

This is a fascinating study on an previously underappreciated component of the wildlife trade. I do wonder how effective patents will be as an early warning for emerging threats to novel wildlife markets. Presumably the number of wildlife species with patents is a very small proportion of all wildlife traded - and a comprehensive understanding of which species are patented and at what stage in their wildlife use/exploitation would be useful. I would like the authors to add some discussion about how emerging patents would be detected for novel species - rather than simply the retrospective analysis as currently provided.

Response

This is a great point. The method we demonstrate here could be adapted for any groups of species, trade names or wildlife products. In theory, it could be set up to detect patents across any number of species for significant shifts in patenting rates, which could in turn be categorised to identify emerging applications.

In addition, we could take a different approach by analysing all patents related to a use type, such as Traditional Chinese Medicine, to look for the emergence of new species being used in these products. With some work to identify the right keywords for these approaches, either could detect emerging increases in patenting for novel species in trade.

We have added this to the paper:

L235 onwards now reads *“For identifying emerging threats more broadly, there are two potential extensions to these approaches. Firstly, automated searches could be conducted for a wide range of species, identified via some method of prioritization, for example, the IUCN Red List or expert advice on taxa that may be important to monitor.*

Then, if increased rates of patenting were detected, these could be investigated in greater detail, to characterize patent types and emerging markets. Alternatively, searches based on keywords that characterise particular classes of patents, such as Traditional Chinese Medicine, could be monitored for the emergence of new species in patent documents. This would reveal the potential for the inclusion of previously unexploited taxa in new products and applications. Of particular importance for the detection of emerging patenting behaviour would be periods following major shocks that affect entrepreneurship (e.g., through opening new opportunities, closing existing markets, or changing consumer attitudes and behaviour), such as changes in regulatory regimes or global events, such as the COVID-19 pandemic.”

Comment 2

Reference 14 would be better as: Heinrich, S., et al., 2019. Of cowboys, fish, and pangolins: US trade in exotic leather. *Conservation Science and Practice*, 1(8), p.e75.

Response

Many thanks – we have replaced the reference as suggested and also added in detail of this paper to highlight its findings, which we agree much better reflect our points here:

L37 onwards now reads *“Furthermore, trade bans or other regulatory events may lead to the emergence of new substitute species from different regions, such as arapaima leather as a replacement for pangolin leather in the USA¹⁴, or increased trade of African lion and South American jaguar products in Asian markets following the international commercial trade ban for tigers^{15,16}”*

L249 onwards now reads *“Past research has shown that the impact of wildlife trade restrictions is often unpredictable, with CITES listings increasing trade in the short term¹¹, and rapidly shifting exploitation to other, less regulated species¹⁴”*

Comment 3

Line 40. 'Non-wild' is an unclear term, are you referring to both farmed and artificial? It would be good to better define this.

Response

We were using 'non-wild' as a catchall term to cover farmed, cultivated, and synthetic products, to save spelling this out each time – however we do agree that this presented some confusion, so we have now changed this to be more specific at each point where it was used.

For example we now say “Farmed, artificially propagated, or lab-grown versions” at first mention of this at L41, and some variation of ‘farmed or synthetic alternatives’ at Lines 139, 172, 208, 213 and 281.

Comment 4

Line 154. Given that 'the number of new patents related to established uses, such as medicines using bear bile, continued to exceed those for novel product types' - how do the authors consider their general findings will apply to novel less consumed/marketable species?

Response

Many thanks – this is a great point. Wildlife trade is so complex that we do not want to speculate about how what we observe here could apply to a novel species in trade. However, we do think it is important to note where and how this method could be used for wider species. We have elaborated on this in our response to comment 1 for currently less consumed species that might emerge in the trade. In addition, we now note in the text that not all wildlife trade will be patented but that this method should be used as a new tool to provide data that could triangulate with other existing data sources.

L289 now reads “We note that not all traded wildlife will be the subject of patenting, but patent data should be triangulated with other data sources to add an additional layer of information about commercial trade in certain markets that has so far been overlooked”

Reviewer #3 (Remarks to the Author):

Comment 1

I enjoyed this paper a lot, and learned from it. Thank you for that.
I have very minor suggestions:

Response

Many thanks!

Comment 2

Ln 43: owl trade in Indonesia due to their rising popularity as pets following the Harry Potter films¹⁸.

Check the validity of this claim, as I have read that it is controversial and more anecdotal than anything else.

Response

We agree with this point, and we are aware of the work by Verissimo et al. who showed no effect of the films on owl demand using robust methods. We originally used this paper here, not to show the purported increases in demand (which are disputed by the paper mentioned above), but more the behaviour of businesses who try adapt to trends by advertising owls as ‘Harry potter birds’. However, we agree that the text as written does not reflect our point, and that the owl example in itself is anecdotal. We have therefore changed this to cite an example of medicinal plants being marketed as COVID-19 cures.

L43 now reads “Furthermore, businesses strategically adapt to shifting consumer or societal trends in product preferences, such as rapid shifts in the marketing of existing medicinal plant products to promote them as COVID-19 cures in early 2020 (e.g. *Artemisia annua* in Madagascar¹⁸)”

And we have added reference to a news story about the rapid branding of this product as a COVID treatment: Rabary, L. (2020). Madagascar coronavirus herbal mix draws demand from across Africa despite WHO misgivings. Reuters. May 8, 2020. <https://www.reuters.com/article/us-health-coronavirus-madagascar-idUSKBN22K1HQ/>

Comment 3

Ln 56: remove last comma

Response

We have now removed both commas on this line (new L58)

Comment 4

Ln 131: For me, this is the most important finding of the paper and warrants more discussion and clarification. At least I would like to read more on this trend.

Response

We found this trend surprising and interesting too. Within the word limits, we have a lot of discussion around the different cases in which this happened in our data, and what could have resulted in this trend (See L145, L248, L264), and we also demonstrate the correlation between patenting and trade bans in Figure 2 (and supporting materials)

However, we agree that more should be said here and we have added some new text to highlight this as a major priority for future research and monitoring:

L276 “Scanning the patent data for species that have recently been subject to trade bans or other regulations to restrict trade could underpin targeted work with businesses or sectors to determine the reasons why they want to patent an illegal product, the best interventions to ensure that this commercial interest is sustainable.”